# Lightwave-driven scanning tunnelling spectroscopy of atomically precise graphene nanoribbons

S. E. Ammerman [1,5], V. Jelic [1,5], Y. Wei [1], V. N. Breslin[1], M. Hassan[1], N. Everett[1], S. Lee[1], Q. Sun [2,4], C. A. Pignedoli [2], P. Ruffieux [2], R. Fasel [2,3] & T. L. Cocker [1✉]

Atomically precise electronics operating at optical frequencies require tools that can characterize them on their intrinsic length and time scales to guide device design. Lightwave-driven scanning tunnelling microscopy is a promising technique towards this purpose. It achieves simultaneous sub-ångström and sub-picosecond spatio-temporal resolution through ultrafast coherent control by single-cycle field transients that are coupled to the scanning probe tip from free space. Here, we utilize lightwave-driven terahertz scanning tunnelling microscopy and spectroscopy to investigate atomically precise seven-atom-wide armchair graphene nanoribbons on a gold surface at ultralow tip heights, unveiling highly localized wavefunctions that are inaccessible by conventional scanning tunnelling microscopy. Tomographic imaging of their electron densities reveals vertical decays that depend sensitively on wavefunction and lateral position. Lightwave-driven scanning tunnelling spectroscopy on the ångström scale paves the way for ultrafast measurements of wavefunction dynamics in atomically precise nanostructures and future optoelectronic devices based on locally tailored electronic properties.

[1] Department of Physics and Astronomy, Michigan State University, East Lansing, MI 48824, USA. [2] Empa, Swiss Federal Laboratories for Materials Science and Technology, 8600 Dübendorf, Switzerland. [3] Department of Chemistry, Biochemistry and Pharmaceutical Sciences, University of Bern, 3012 Bern, Switzerland. [4]Present address: Materials Genome Institute, Shanghai University, 200444 Shanghai, China. [5]These authors contributed equally: S. E. Ammerman and V. Jelic. ✉email: cockerty@msu.edu

Scanning probe microscopy drives progress in materials science through atomically resolved real-space imaging of new compounds and nanostructures[1,2]. Yet, its success is not due to atomic spatial resolution alone. An equally important aspect is that scanning probe microscopy techniques can extract rich spectroscopic information at each spatial position. For example, combining scanning tunnelling microscopy (STM) and spectroscopy (STS) in differential conductance (d$I$/d$V$) maps reveals a sample's local density of electronic states (LDOS), i.e., as a function of both position and energy. Scanning probe techniques that visualize orthogonal material properties on the atomic scale are also emerging, including electro- and photoluminescence STM[3,4], electron spin resonance STM[5], and tip-enhanced Raman microscopy[6,7]. Lightwave-driven STM[8–21] opens another dimension: atomically resolved microscopy on ultrafast timescales. It has enabled femtosecond pump-probe experiments[8,9,13,19], field-driven electroluminescence[20], and the application of atomic-scale ultrafast forces[15,17]. Ultrafast fields can also operate in regimes inaccessible to conventional static STM fields, for example through lightwave-control of extreme tunnel currents[10–12,16,21]. The natural next step is to bring these capabilities to bear on novel material systems to inform design and synthesis. However, atomically resolved lightwave-driven spectroscopy is needed to realize this promise, and its demonstration remains outstanding.

Here, we investigate the promising material platform of atomically precise graphene nanoribbons[22] (GNRs) on the ångström scale with lightwave-driven terahertz scanning tunnelling microscopy (THz-STM) and spectroscopy (THz-STS). The small duty cycle of lightwave-driven tunnelling facilitates measurements at ultralow tip heights, where the GNR electronic wavefunctions feature far richer spatial structure than at conventional STM tip heights. THz-STS performed as a function of three-dimensional position above the GNR allows us to extract the differential conductance sampled by lightwave-driven tunnelling with ångström horizontal and sub-ångström vertical resolution. Our measurements show that the vertical decay of the differential conductance is remarkably sensitive to lateral position and electronic orbital. Inspired by this observation, we introduce lightwave-driven scanning tunnelling tomography, where constant-height THz-STM images illustrate a transition from tunnelling dominated by occupied states in the GNR valence bands to tunnelling dominated by unoccupied states in the GNR conduction bands within a tip-sample distance increase of just 1 Å.

## Results

**Lightwave-driven microscopy at ultralow tip heights**. The graphene nanoribbons are grown from molecular precursors[23] on a clean Au(111) substrate by on-surface synthesis[22] (see Methods). We select 7-atom-wide GNRs with armchair edges (7-AGNRs) for the current study, as depicted in Fig. 1a. For conventional STM imaging, the tip height $z = z_0$ is defined by the bias voltage $V_{d.c.}$ and current setpoint $I_{d.c.}$. Conventional STS is performed by disengaging the feedback loop, applying an additional sinusoidal voltage of perturbative amplitude d$V$ (e.g., with millisecond oscillation period) and detecting the sinusoidal current amplitude d$I$ as a function of $V_{d.c.}$ (Fig. 1b). This yields the slope of the current–voltage characteristic (i.e., d$I$/d$V$), which is approximately proportional to the energy-dependent LDOS of the sample (see Methods).

In contrast, in THz-STM and THz-STS, the terahertz voltage probe is nine orders of magnitude faster than conventional voltage modulation and only a single oscillation cycle long. As the terahertz pulses are generated by an ultrafast near-infrared laser (see Methods and Supplementary Figure 1), a train of terahertz

pulses is delivered to the junction. Each terahertz pulse stimulates a sub-picosecond rectified current pulse through lightwave-driven tunnelling, thereby producing a train of current pulses with a duty cycle equal to the current-pulse duration divided by the time between terahertz pulses (~1 ps / 1 μs = $10^{-6}$). Only the average terahertz-induced current can be measured, so the peak current must compensate for the duty cycle by far exceeding $I_{d.c.}$, as summarized in Supplementary Figure 2. Consequently, the conventional method of STS is incompatible with the key aspect of lightwave-driven tunnelling that makes ultrafast time resolution possible, and an alternate approach is required for THz-STS. Here, THz-STS spectra are recorded by sweeping the peak field strength of the incident terahertz pulse while maintaining a uniform waveform shape, then inverting the polarity and repeating[10,11,13,16].

In the THz-STM and THz-STS experiments reported here, the initial tip height is set using the conventional STM feedback loop. The feedback loop is then disengaged, the bias is reduced to 0 V, and the tip is approached 3–4 Å closer for lightwave-driven imaging and spectroscopy, e.g., to $z = z_0 - 4$ Å (Fig. 1c). On clean surfaces the tunnelling rate increases by one order of magnitude for each ångström of approach, effectively rescaling the current–voltage characteristic (Fig. 1d). This not only compensates for the ~$10^{-6}$ duty cycle of lightwave-driven tunnelling, but also enables measurements at ultralow tip heights. We define ultralow tip heights as distances at which orbitally selective imaging by conventional STM with an s-wave tip is unfeasible because the d.c. current would either damage the tip apex or surface under investigation or lead to the tip picking up the GNR[24]. Conversely, lightwave-driven STM is possible because sub-picosecond current pulses of the same peak magnitude consist of only a few electrons.

In practice, single-cycle, phase-stable terahertz pulses are coupled through free space to the STM tip apex, where they act as ultrafast voltage transients on the junction's current–voltage characteristic, as depicted in Figs. 1c and 1d. The temporal integral of the lightwave-driven current, i.e., the rectified charge $Q_{THz}$, is registered through a shift in the average current of $I_{THz} = f \times Q_{THz} = f \times e \times N_e$, where $e$ is the elementary charge, $f$ is the repetition rate of the amplified near-infrared laser system used to generate the terahertz-pulse train, and $N_e$ is the average number of rectified elementary charges per terahertz pulse. Notably, the peak of the ultrafast terahertz-induced current, $i_{THz,pk}$, can reach the microamp scale within a sub-picosecond oscillation cycle (Fig. 1d).

The chemical structure of the 7-AGNR is depicted in Fig. 2a. Differential conductance mapping of the 7-AGNR valence band in constant-height mode reveals that the LDOS is primarily located outside the atomic frame at a tip height of $z_0$ (Fig. 2b), consistent with previous observations[25,26]. For comparison, we calculate the 7-AGNR electron density as a function of energy and height above the atomic plane (see Supplementary Figure 3) using density functional theory (DFT). The DFT tomographic cut through the valence band at $z = 5$ Å agrees with the d$I$/d$V$ map (compare Figs. 2b and c), as has been demonstrated previously[26], where the DFT height is lower than the experimental tip height due to an artifact of the exchange-correlation functional and basis set approximations adopted in DFT. Conversely, the DFT cut through the valence band at lower height, e.g., at $z = 2$ Å as shown in Fig. 2d, predicts the spatial structure probed in lightwave-driven experiments. The image is notably richer in detail, since the wavefunction in the interior of the 7-AGNR is composed of lobes of alternating sign that destructively interfere at higher tip heights[26]. These lobes influence STM images recorded with CO functionalized tips[27], but the mixed s- and p-wave character of the tip makes the corresponding microscopy

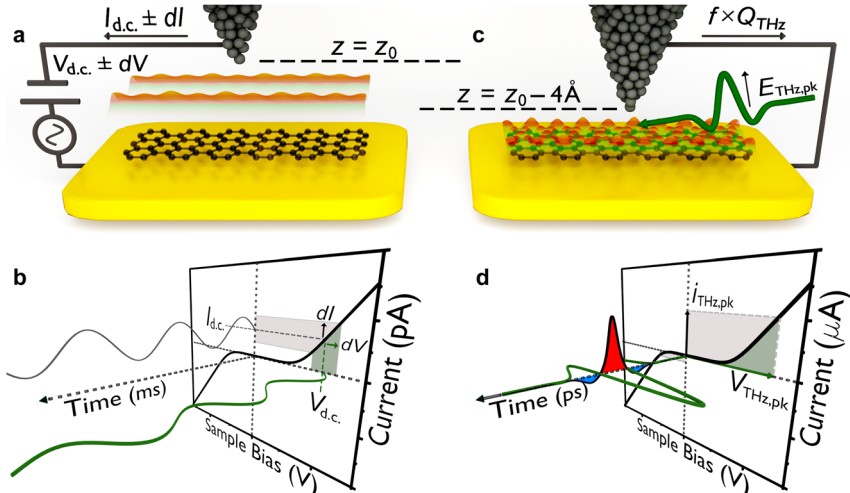

**Fig. 1 Comparing ultrafast terahertz scanning tunnelling spectroscopy (THz-STS) to conventional STS. a** Schematic diagram of the tunnel junction during scanning tunnelling microscopy (STM) and STS of a seven-atom-wide armchair graphene nanoribbon (7-AGNR). The tip height, $z_0$, is set by the tunnel current feedback conditions, e.g., with steady-state bias voltage $V_{d.c.} = -1$ V and steady-state tunnel current setpoint $I_{d.c.} = -50$ pA. **b** Schematic current–voltage ($I$–$V$) characteristic of a 7-AGNR (solid black curve) with the time axis of the millisecond sine-wave modulation for conventional STS projected out-of-plane. In STS, the steady-state bias voltage is modulated by a sine-wave of amplitude $dV$ (solid green curve), generating a tunnel current modulation of amplitude $dI$ (solid grey curve). **c**, **d** In THz-STM and THz-STS, a phase-stable single-cycle terahertz pulse with a peak electric field of $E_{THz,pk}$ is coupled through free space to the STM tip apex, where it acts as an ultrafast transient bias with a peak voltage of $V_{THz,pk}$. The tip height is reduced by approximately 4 Å to account for the reduced duty cycle of the bias compared to conventional STM. At this reduced tip height, the 7-AGNR local density of states (LDOS) features far richer spatial structure than at STM and STS tip heights (orange and green overlays in **a** and **c**). A terahertz voltage transient induces a picosecond current pulse, $i_{THz}(t)$, defined by the $I$–$V$ characteristic, with peak current, $i_{THz,pk}$, reaching the µA scale, compensating for the low duty cycle of THz-STS. The average rectified component of the current pulses, $Q_{THz} = eN_e = \int i_{THz}(t)dt$, is measured as a steady-state current, $I_{THz} = f \times e \times N_e$, where $e$ is the elementary charge, $f$ is the repetition rate of the laser and $N_e$ is the average number of rectified elementary charges per terahertz pulse. The red and blue shadings underneath the picosecond current transient in **d** emphasize the rectification process for a single pulse. The green and tan shaded regions in **b** and **d** highlight the range of the voltage modulation and resulting instantaneous tunnel current.

and spectroscopy difficult to interpret[28], while conventional STM with an $s$-wave tip cannot access the necessary ultralow tip heights. In general, lowering the bias allows the tip to be moved closer (e.g., as is the case for bond-resolved STM using a CO functionalized tip[29]) but the tunnelling electrons are not resonant with the molecular orbital LDOS at low bias and the in-gap LDOS is probed instead.

Interpreting lightwave-driven images can also be nontrivial, since the voltage transient sweeps through both polarities, acting on all LDOS features in the voltage window defined by $V_{THz,pk}$ and the waveform symmetry (Fig. 1d). Precisely characterizing the temporal structure of the voltage waveform at the tip apex is therefore essential. Electro-optic sampling in the far-field provides the terahertz electric field transient focused onto the STM tip (Figs. 2e and f), but antenna effects[11,14,17] and geometrical features on the scale of the terahertz wavelength[12,13,16,17] can modify its shape. Therefore, in-situ characterization of the near-field waveform is needed[13,14,17,30], e.g., photoemission sampling (PES, see Supplementary Fig. 4 for further details). As terahertz field enhancement is affected by the microscopic tip profile[17,21]—which must be carefully prepared over the substrate for high-resolution THz-STM imaging—we use PES to measure the waveform shape but determine the terahertz voltage calibration from THz-STM and THz-STS. We select between an even or odd number of terahertz-pulse reflections before coupling the pulses to the tip, and hence invert the polarity of the terahertz voltage transient at the apex without changing its temporal shape or amplitude, as confirmed by PES in Figs. 2e and f. We refer to the electric field transient in which the largest field is negative as $E_{THz,pk} < 0$ (and vice versa), where the field direction is consistent with the STM bias volta.

Figure 2g shows a constant-height THz-STM image of a 7-AGNR recorded for $E_{THz,pk} < 0$ at $z = z_0 - 4$ Å. Remarkably, it is

almost identical to the DFT calculation of the electron density at $z = 2$ Å, $V = -0.9$ V (Fig. 2d), which is dominated by the valence band (see Supplementary Fig. 3), and features exquisite, ångström-scale detail not observable in the conventional differential conductance image (Fig. 2b). The rectified charge is predominantly negative, consistent with lightwave-driven tunnelling of electrons from the GNR to the tip. Conversely, the THz-STM image for $E_{THz,pk} > 0$ is significantly more complicated (Fig. 2h). The anti-nodes of negative rectified charge persist, but the previously nodal regions in the interior of the ribbon are replaced by areas of positive rectified charge, i.e., electrons that have moved from the tip into the sample. The mixture of positive and negative rectified charge in Fig. 2h signifies that the $E_{THz,pk} > 0$ waveform probes considerable sample LDOS on both sides of the Fermi level. The relative weighting between these spectral features depends sensitively on ångström-scale tip position, manifesting as fine local detail in Fig. 2h.

**Lightwave-driven spectroscopy with ångström resolution**. We apply spatially dependent THz-STS to disentangle the intrinsic properties of the 7-AGNR at ultralow tip heights from the lightwave-driven tunnelling process. As a roadmap, conventional $dI/dV$ spectra are recorded for the edge of a 7-AGNR and for the substrate at a tip height of $z_0$ (Fig. 3a). These measurements are consistent with previous reports[25,26]. We then select two characteristic locations for THz-STS (Fig. 3b), which we label the node (green circles in Fig. 3b) and anti-node (red diamonds in Fig. 3b) based on the THz-STM image for $E_{THz,pk} < 0$. The corresponding positions above the atomic structure are determined by comparison with the DFT calculation in Fig. 2d. Figures 3c and d show THz-STS spectra as a function of peak field strength,

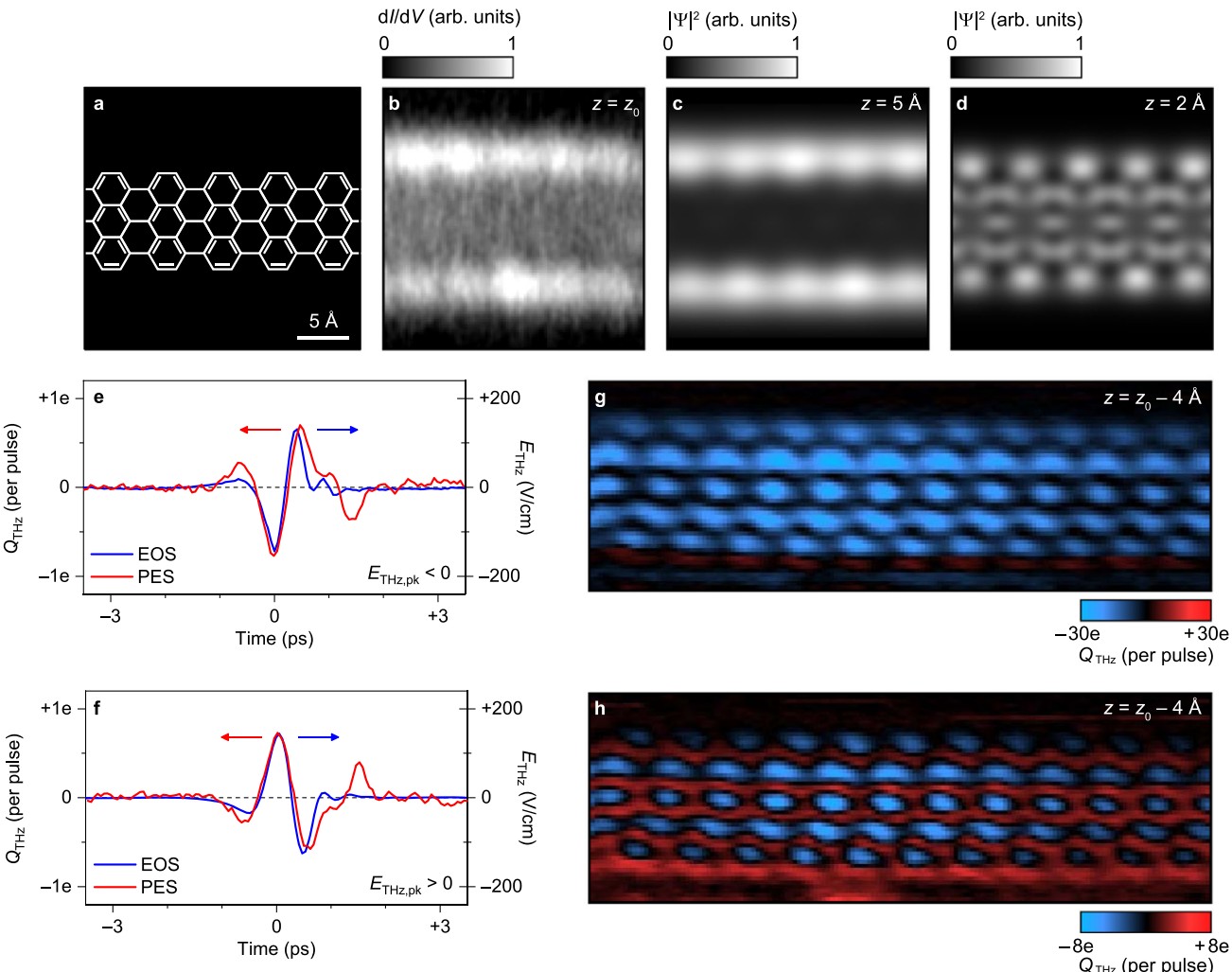

**Fig. 2 THz-STM of seven-atom-wide graphene nanoribbons on Au(111). a** The chemical sketch of a 7-AGNR segment. The scalebar applies to all images in the figure. **b** Measured steady-state STM constant-height differential conductance map (d$I$/d$V$) acquired at $V_{d.c.} = -1$ V. The initial tip height, $z_0$, was set with the tip located at the center of the nanoribbon at $V_{d.c.} = -1$ V and $I_{d.c.} = -50$ pA. **c, d** Electron probability density ($|\Psi|^2$) at $V = -0.9$ V calculated using DFT at a tip height of 5 Å (**c**) and 2 Å (**d**). **e, f** Terahertz-pulse waveforms measured via free-space electro-optic sampling (EOS, blue curves) and photoemission sampling at the tip apex (PES, red curves). The carrier-envelope-phase shift between **e** and **f** is 180°. **g, h** THz-STM constant-height rectified charge maps, $Q_{THz}(x,y)$, acquired at $E_{THz,pk.} = -180$ V/cm (**g**) and $E_{THz,pk.} = +180$ V/cm (**h**), with $V_{d.c.} = 0$ V and $z = z_0 - 4$ Å for both images.

where the rectified charge is measured at the anti-node and node, respectively, for tip heights of $z = z_0 - 4$ Å and $z = z_0 - 3.5$ Å.

To extract the differential conductance sampled by lightwave-driven tunnelling at each tip position, we adopt a model in which the 7-AGNR band onsets at positive and negative voltages are described by error functions and the Au(111) surface state is described as a Gaussian (see Methods). Their energetic positions and widths are constrained by fits to the STS data, as shown in Fig. 3a. We then simulate charge rectification by a voltage pulse matching the terahertz near-field waveform (Figs. 2e and f), allowing the amplitude of the three LDOS contributions to vary until the simulation optimally fits the measured $Q_{THz}$–$E_{THz,pk}$ data (lines in Figs. 3c and d). The terahertz voltage calibration is used as a global fit parameter for all data recorded without a tip change (see Methods). The extracted d$I$/d$V$ curves at the anti-node and node (Figs. 3e and f, respectively) are consistent with THz-STM imaging (Fig. 2g and h).

We next combine THz-STS with THz-STM to extract spatial maps of the d$I$/d$V$ amplitudes. THz-STM images are recorded as a function of $E_{THz,pk}$ for both positive and negative polarity (Supplementary Fig. 5), allowing $Q_{THz}$–$E_{THz,pk}$ curves to be

plotted for every pixel. Applying our model to each pixel individually (with terahertz voltage calibration as a global parameter) yields the extracted d$I$/d$V$ amplitude maps at negative (Fig. 3g) and positive (Fig. 3h) bias. They agree with the corresponding DFT images at $z = 2$ Å, $V = -0.9$ V (Fig. 3i) and $z = 2$ Å, $V = +1.3$ V (Fig. 3j), respectively, which account for the closely spaced[26] valence band (VB) and VB − 1 at negative bias and the conduction band (CB) and CB + 1 at positive bias.

As further corroboration of our model and the fidelity of the PES waveform, and as a test for multielectron and dynamic effects, we perform autocorrelation measurements[8,9,11,12,16,21]. The terahertz pulses are split in half, a delay is introduced, and $Q_{THz}$ is recorded as a function of delay time for three different $E_{THz,pk}$ values at both the anti-node and node (Supplementary Figure 6). A single differential conductance curve is sufficient to reproduce the data at a given tip position (with a constant terahertz voltage calibration), indicating that the d$I$/d$V$ model, combined with the measured PES waveform, sufficiently captures the details of terahertz-induced tunnelling for this system. Finally, we record an additional dataset for a different tip apex and 7-AGNR (Supplementary Fig. 7). In this case, the higher order

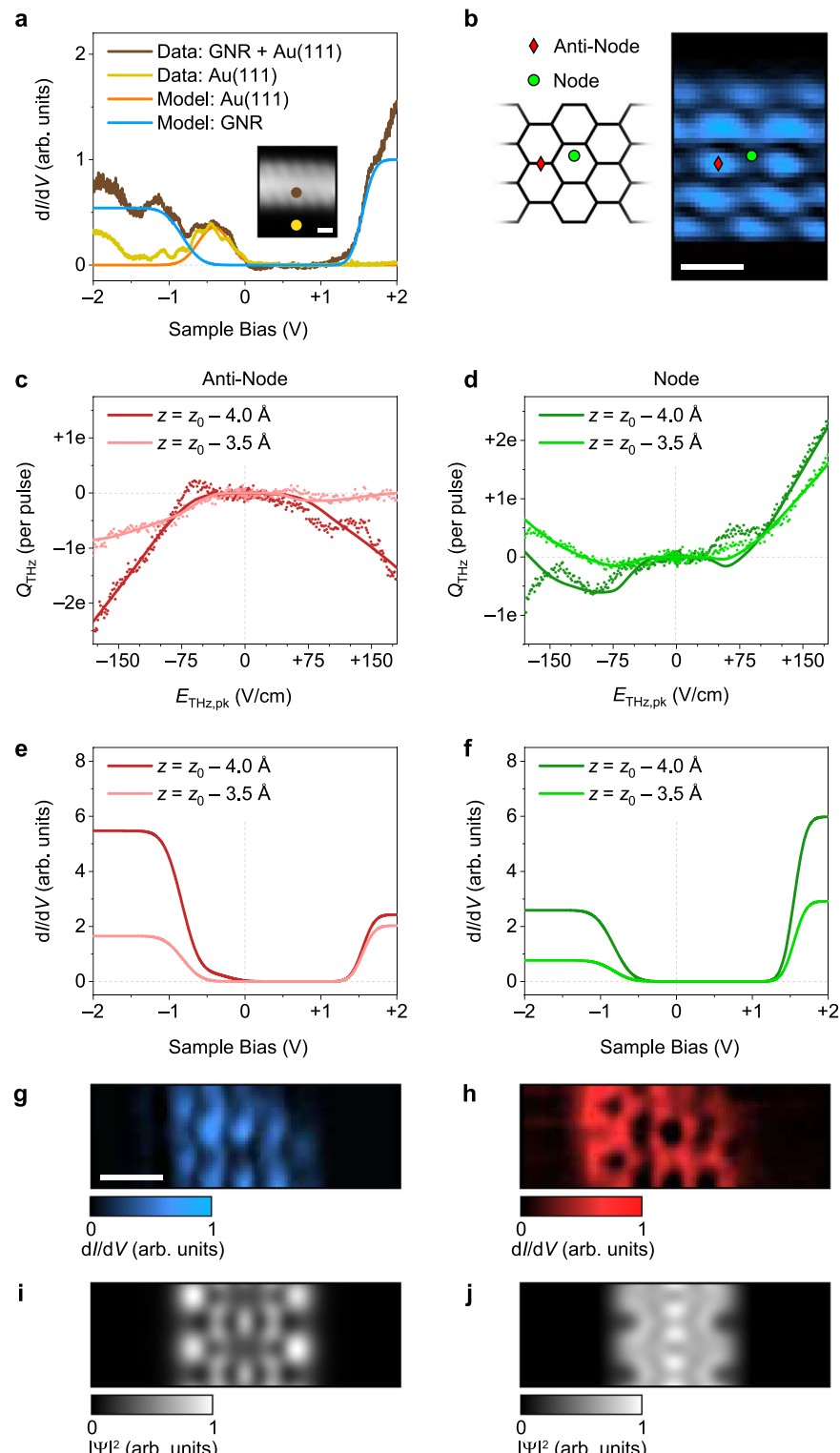

**Fig. 3 Terahertz scanning tunnelling spectroscopy (THz-STS) of a 7-AGNR with ångström-scale resolution. a** Conventional STS of a 7-AGNR adsorbed onto an Au(111) surface (solid brown line) and the Au(111) surface state (solid yellow line). The differential conductance of the gold surface state is modeled by a Gaussian centered at –0.4 V (solid orange line), while the differential conductance of the 7-AGNR is modeled as two error functions near the band edges (solid blue line). Inset: typical steady-state constant-height STM image showing the tip positions for tunnelling spectroscopy. **b** Tip positions for THz-STS overlayed on a diagram of the physical structure of a 7-AGNR (left) and a $Q_{THz}$ map (right). Diamonds: anti-node position; circles: node position. **c, d**, Measured (circles) and simulated (solid lines) $Q_{THz}$–$E_{THz,pk}$ curves at $z = z_0$ - 4 Å (dark lines and symbols) and $z = z_0$ - 3.5 Å (light lines and symbols) with the tip positioned at the anti-node (**c**) and node (**d**); $V_{d.c.} = 0$ V. **e, f**, Model d$I$/d$V$ extracted from the fits in **c** and **d** for the anti-node (**d**) and node (**f**) tip positions, respectively. **g, h** Model d$I$/d$V$ maps at negative (**g**) and positive (**h**) bias extracted pixel-by-pixel from field-dependent $Q_{THz}(x,y)$ images in Supplementary Figure 5. **i, j** Electron probability density ($|\Psi|^2$) at 2 Å and $V$ = –0.9 V (**i**) and $V$ = +1.3 V (**j**) calculated by DFT. The dashed lines in **c**–**f** are guides to the eye. The scalebars in **b**, **g**, and the inset of **a** are 5 Å. The scalebar in **g** applies to **g**–**j**.

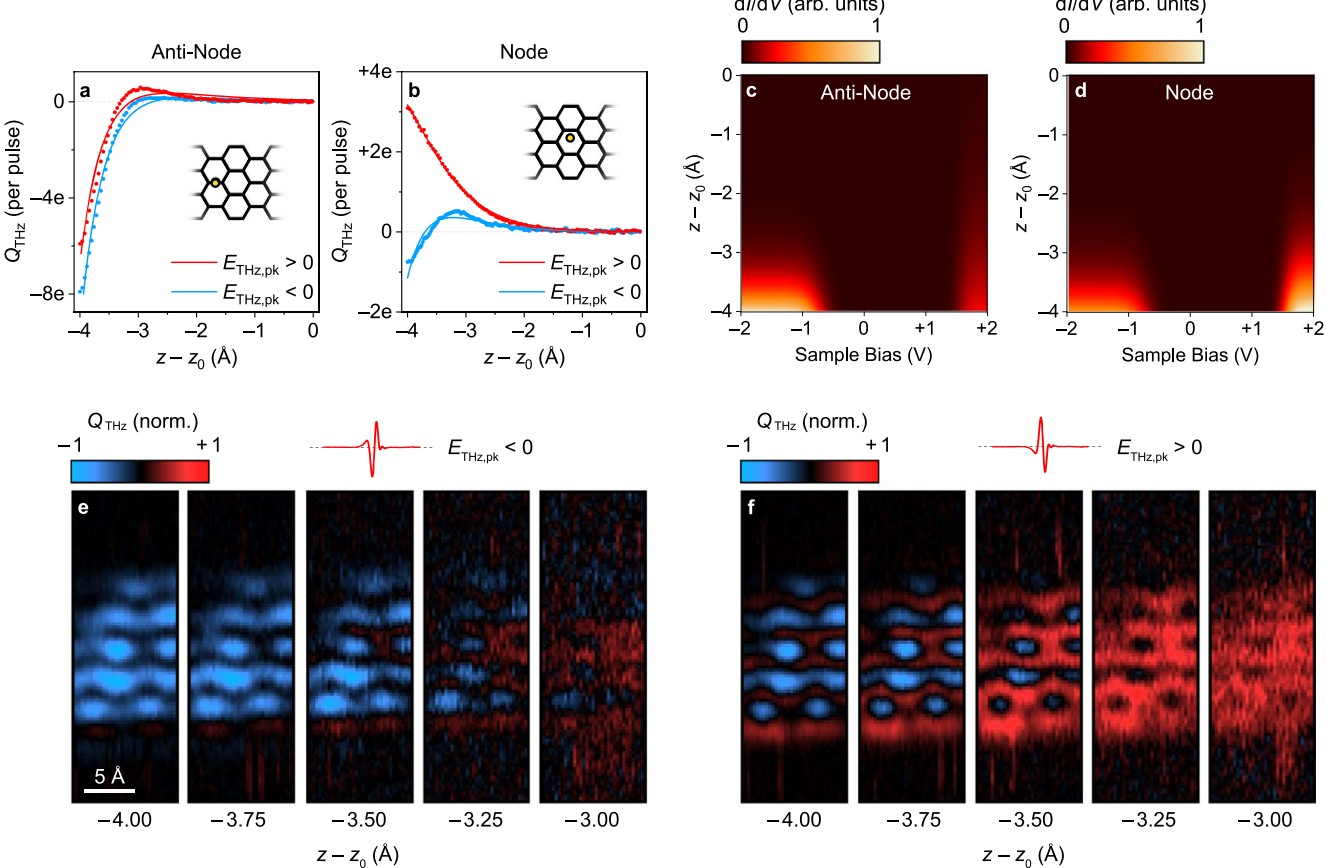

**Fig. 4 Tip height-dependent THz-STM of a 7-AGNR on Au(111). a, b** Measured (circles) and simulated (solid lines) tip retraction scans at the anti-node (**a**) and node (**b**) positions (see inset) acquired at $V_{d.c.} = 0$ V, $E_{THz,pk} = -180$ V/cm (blue) and $E_{THz,pk} = +180$ V/cm (red). **c, d** Differential conductance at the anti-node (**c**) and node (**d**) spatial positions for tip heights, $z$, ranging from $z_0$ to $z_0 - 4$ Å extracted from the fits in **a** and **b**, respectively. The orbital decays are modeled using the expression $e^{-2\kappa z}$. Negative bias fit parameters: $\kappa = 1.3$ Å$^{-1}$ at the anti-node and $\kappa = 1.2$ Å$^{-1}$ at the node. Positive bias fit parameters: $\kappa = 0.4$ Å$^{-1}$ at the anti-node and $\kappa = 0.9$ Å$^{-1}$ at the node. **e, f** Lightwave-driven scanning tunnelling tomography, i.e., $Q_{THz}(x,y)$ vs. $z$, acquired at $E_{THz,pk} = -180$ V/cm (**e**) and $E_{THz,pk} = +180$ V/cm (**f**) with $V_{d.c.} = 0$ V. Scalebar in **e** applies to all panels in **e** and **f**.

oscillations in the THz-STS data are absent and the agreement with the model is even better.

Across all datasets, the extracted $dI/dV$ curves at the anti-node and node are significantly different from one another, even though they are laterally separated by only ~2 Å, a distance comparable to the carbon-carbon bond length (Fig. 3b). Retracting the tip by just 0.5 Å at either position (i.e. from $z = z_0 - 4$ Å to $z = z_0 - 3.5$ Å) leads to a drastic change to both the measured $Q_{THz}-E_{THz,pk}$ curve (Figs. 3c and d) and the extracted $dI/dV$ (Figs. 3e and f, respectively). Our measurements show spatially dependent THz-STS on the ångström scale and illuminate a rich spatial structure to the LDOS of 7-AGNRs at ultralow tip heights. Most remarkably, we observe that the amplitude of the extracted $dI/dV$ at the anti-node position changes far more at negative voltages than positive voltages with increasing tip-sample distance (Fig. 3e). In contrast, at the node position, the $z$-dependent change to the $dI/dV$ at positive and negative voltages is similar (Fig. 3f). Hence, at ultralow tip heights, the vertical decays of the wavefunctions at positive and negative bias depend sensitively on in-plane position and can be significantly different from one another.

**Lightwave-driven tomography of local wavefunction decay.** We explore the vertical decays of the 7-AGNR wavefunctions in further detail by recording the rectified charge as a function of tip height above the anti-node (Fig. 4a) and node (Fig. 4b). For each

position, the $Q_{THz}-z$ measurement is repeated for terahertz pulses of each polarity with identical $|E_{THz,pk}|$. We apply our model to fit the $Q_{THz}-z$ data, with the $z$ dependence encompassed by an independent exponential decay for each of the three $dI/dV$ features (i.e. the error functions at positive and negative bias and the gold surface state). Measurements recorded with positive and negative $E_{THz,pk}$ for a given position are fit simultaneously, yielding $dI/dV$ as a function of height at the anti-node (Fig. 4c) and node (Fig. 4d). We find that the decay of the error function at negative bias is consistently faster than that at positive bias. This is partially explained by different tunnelling barrier heights, since the onset of the LDOS is closer to the Fermi level at negative bias (Fig. 3a). The dependence of the vertical decay rate on horizontal tip position, meanwhile, can be traced back to the complex spatial structure of the wavefunctions at ultralow tip heights. Interference between the probability amplitude lobes of positive and negative sign in an individual wavefunction introduces an additional vertical decay to the LDOS that is wavefunction specific[26] (Supplementary Figure 3). This effect is not observed with conventional STM because destructive interference between the lobes is complete at standard tip heights.

Whereas DFT calculations of the LDOS at constant-height agree with our THz-STM and THz-STS measurements (e.g., Figure 3g–j), the vertical LDOS decays predicted by DFT are not reliable due to the fast decay of the Gaussian basis set and limitations of the exchange-correlation functional. Therefore, we

compare the experimental decays to tight-binding calculations (Supplementary Fig. 8). Unlike our DFT calculations, the tight-binding approach considers the wavefunctions separately and does not account for the gold substrate. Nevertheless, reasonable agreement is obtained, albeit under the assumption that each experimental decay can be associated with a particular band (VB − 1, VB, CB, or CB + 1) that dominates the decay at that tip position and energy.

Interestingly, the $Q_{THz}$–z decay curves for $E_{THz,pk} > 0$ at the anti-node and $E_{THz,pk} < 0$ at the node feature prominent sign changes between $z = z_0 − 4$ Å and $z = z_0 − 3$ Å. These sign changes are explained by a competition between the asymmetry of the terahertz waveform (Figs. 2e and f) and the relative strengths of the valence and conduction bands in the height-dependent d$I$/d$V$ maps (Figs. 4c and 4d). We investigate the spatial dependence of this competition further with constant-height THz-STM images as a function of tip height, $Q_{THz}(x,y)$ vs. $z$, which we call lightwave-driven scanning tunnelling tomography. Figure 4e shows tomography images for $E_{THz,pk} < 0$, while Fig. 4f shows the complementary dataset for $E_{THz,pk} > 0$. The transition from $Q_{THz} < 0$ to $Q_{THz} > 0$ is most evident in Fig. 4f, where the asymmetry of the terahertz voltage waveform compensates for the asymmetry of the LDOS onsets relative to the Fermi level. In Fig. 4f, the THz-STM image at $z = z_0 − 4$ Å largely matches the spatial structure observed for the opposite polarity (Fig. 4e at $z = z_0 − 4$ Å), albeit with positive rectified charge at the nodal positions. By $z = z_0 − 3.25$ Å, the THz-STM image is entirely composed of positive rectified charge (Fig. 4f, right).

To complete our tomography analysis, we assemble $Q_{THz}$–z curves for every pixel in Fig. 4e, f and apply our model, as in Fig. 4a, b, to reveal two-dimensional cross-sections through the differential conductance as a function of tip height at positive and negative bias (Supplementary Fig. 9). In addition to these tomographic reconstructions, we also extract the spatial dependence of the vertical decay parameters with ångström in-plane resolution (Supplementary Fig. 9). These maps are consistent with our point-spectroscopy analysis in Fig. 4a–d.

## Discussion

Looking ahead, lightwave-driven scanning tunnelling microscopy, spectroscopy, and tomography of atomically precise GNRs sets the stage for a new era of ångström-scale ultrafast science in novel material systems. On-surface[22] and solution synthesis[31] of GNRs enable wide-ranging design possibilities, including doping, edge functionality, and even topological electronic states[32,33]. With pump-probe THz-STM and THz-STS, the optoelectronic properties[34–38] of individual GNRs will soon come into clearer view, visualized through atomically resolved movies of sub-picosecond population[39,40] and wavefunction[41] dynamics. Our terahertz-pulse autocorrelation measurements (Supplementary Fig. 6) indicate that a dielectric decoupling layer will likely be needed to unveil the intrinsic GNR dynamics, but established manipulation of GNRs with the STM tip[42] provides a clear roadmap for constructing optimized protypes of atomically precise optoelectronics.

## Methods

**Terahertz scanning tunnelling microscope**. The STM is a custom-designed, commercial ultrahigh vacuum (UHV) low-temperature STM system with a base pressure of $5 \times 10^{-11}$ mBar (CreaTec Fischer & Co. GmbH). The STM tip is electrochemically etched from 0.35 mm diameter polycrystalline tungsten wire. The tips were cleaned in situ using electron bombardment heating and prepared with field-directed-sputter-sharpening[43]. The experiments were performed at a tip and sample temperature of 77 K. The bias voltage is applied to the sample and the tunnel current is detected using a preamplifier (Femto DLPCA-200) with a gain of $10^9$ and a bandwidth of 1 kHz. Constant height THz-STM was performed at zero

bias ($V_{d.c.} = 0$ V), with the number of elementary charges rectified by each THz pulse, $N_e$, determined via $N_e = (I_{THz})/(e \times f)$, where $f = 1$ MHz is the repetition rate and $e$ is the elementary charge. The THz pulse enters the UHV chamber through a sapphire viewport and propagates through two additional sapphire windows on the heat shields before being focused onto the tip by a 60° off-axis polished aluminum parabolic mirror with a focal length of 33.85 mm and a diameter of 25.4 mm. The position of the parabolic mirror is fixed relative to the position of the scanning tip. The linearly polarized THz pulse is coupled to the STM tip at an incident angle of 30° with the THz polarization along the tip axis (p-polarized with respect to the sample).

**Sample preparation**. A single crystal Au(111) sample was used as the substrate for the growth of 7-AGNRs[23]. The Au(111) surface was cleaned by repeated cycles of argon ion bombardment and annealing to 850 K. For growth of 7-AGNRs, 10,10'-dibromo-9,9'-bianthryl precursor molecules were deposited onto the clean Au(111) surface via sublimation from a Knudsen cell at a rate of ~10 Å/min. The substrate temperature was raised from 300 to 463 K over 30 min to induce dehalogenation and then held at 463 K for 10 min to facilitate polymerization of precursor molecules. A final annealing step was performed for 10 min at 663 K to allow for cyclodehydrogenation of the polymers into 7-AGNRs.

**Ultrafast laser source and terahertz optics**. The laser system is a ytterbium-doped potassium gadolinium tungstate (Yb:KGW) regenerative amplifier with a repetition rate of 1 MHz, center wavelength of 1030 nm, and pulse duration of 230 fs at full-width at half-maximum (Light Conversion CARBIDE). Single-cycle phase-stable THz pulses are generated in lithium niobate via tilted-pulse-front optical rectification[44] of 18 µJ near-infrared femtosecond laser pulses. The electric field amplitude of the generated THz pulse is scaled without changing the waveform shape by a pair of wire-grid-polarizers (for $E_{THz,pk} > 5\%$ of maximum), while the field can be inverted using a flip-mirror that modifies the THz beam path by one reflection (the carrier-envelope-phase changes by 180°; see Supplementary Fig. 1). The THz pulse is sent through a Michelson interferometer in which the path length of one arm is controlled by a motorized linear stage. Coherent detection of the free-space THz pulse electric field is performed via electro-optic sampling in a 100-µm-thick ZnTe(110) crystal using a near-infrared gate pulse with a center wavelength of 800 nm and a pulse duration of 10 fs. The 100-µm-thick ZnTe(110) crystal is uniformly bonded to a 2-mm-thick ZnTe(100) in order to temporally delay the reflected THz pulse within the thin crystal to well beyond the field profile of the main THz transient. The broadband spectrum of the gate pulse is produced through supercontinuum generation of 0.5 µJ near-infrared pulses from the regenerative amplifier in a 4-mm-thick yttrium aluminum garnet crystal. The gate pulse is then spectrally filtered in the Fourier plane and compressed to a 10 fs duration (full-width at half-maximum). A mechanical chopper was used to modulate the THz generation pulse train for lock-in detection of the electro-optic signal and the THz pulse-induced current. The THz pulse generation and THz pulse propagation to the STM tip were all enclosed in a dry-air environment with less than 0.1% humidity in order to eliminate water-vapor-induced oscillations in the THz waveform.

**Terahertz electric field calibration**. The signal intensity difference in a pair of balanced photodiodes ($A$, $B$) is used to calibrate the THz pulse electric field via[45]

$$E_{THz,pk} = \frac{1}{t_{ZnTe}} \left( \frac{A - B}{A + B} \right) \left( \frac{\lambda}{2\pi n_O^3 r_{41} L} \right) \quad (1)$$

where $\lambda = 800$ nm is the center wavelength of the gate pulse, $n_O = 2.853$ is the optical refractive index for ZnTe, $r_{41} = 4.04 \times 10^{-12}$ m/V is the electro-optic coefficient for ZnTe, $L = 100$ µm is the crystal thickness and $t_{ZnTe} = 0.48$ is the Fresnel amplitude transmission coefficient for a THz pulse incident onto ZnTe[45]. For THz pulses focused onto the STM tip, the quoted value of the free-space THz pulse electric field also incorporates the Fresnel amplitude transmission coefficient through three sapphire windows on the STM heat shields ($t_{window}^3 = 0.60^3$, as measured by THz transmission spectroscopy with our source), one reflection from an indium-tin-oxide-coated window ($r_{ITO} = 0.90$), and seven reflections while propagating to the STM tip ($r_{mirror}^7 = 0.99^7$). Finally, the quoted THz pulse electric field is scaled to account for the different focusing conditions that occur at the ZnTe crystal and STM tip. Specifically, we multiply the THz pulse electric field by the $f$-number ratio, $f_{ZnTe}/f_{STM}$, for the two different geometries ($f_{STM} = 33.85$ mm / 25.4 mm; $f_{ZnTe} = 50.80$ mm / 25.4 mm).

**Ultrafast photoemission THz near-field sampling**. Broadband phase-resolved detection of the THz pulse temporal profile at the tip apex was performed using ultrafast photoemission sampling[13,14] (PES; Supplementary Fig. 3a). A 230 fs duration (full-width half-maximum) photoexcitation pulse was picked off from the regenerative amplifier laser system and frequency doubled in a β-BBO crystal to a center wavelength of $\lambda = 515$ nm before being sent to the STM tip (see Supplementary Fig. 1). The photoexcitation pulse and THz pulse propagate collinearly to the STM tip. For PES, the tip is retracted by ~1 µm from the sample surface with $V_{d.c.} = +10$ V. At this bias voltage, the current–voltage characteristic of the STM junction is linear (Supplementary Fig. 3b), enabling direct mapping of the

near-field waveform (Supplementary Fig. 3c) and spectrum (Supplementary Fig. 3d) at the tip apex. The photoexcitation pulse is sufficiently short to omit convolution effects when sampling the near-field THz waveform[46]. The measured asymmetry between the main peaks of the terahertz waveform is approximately 20%. If no asymmetry were present, deconvoluting the THz-STS data would be more challenging, if not impossible.

**Computational methods.** We performed density functional theory calculations to obtain the equilibrium geometry of a 7-AGNR. Gas phase calculations provide the wavefunctions shown in Supplementary Figure 3a. Calculations for a 7-AGNR adsorbed on Au(111) provide the electron probability density maps shown in Figs. 2c, 2d, 3i, 3j, and S3b. We used the CP2K code[47] and the AiiDAlab platform[48]. The electronic states were expanded with a TZV2P Gaussian basis set[49] for carbon and hydrogen atoms and a DZVP basis set for gold atoms. A cutoff of 600 Ry was used for the plane wave basis set. We used norm-conserving Goedecker-Teter-Hutter pseudopotentials[50] and the Perdew–Burke–Ernzerhof exchange-correlation functional[51] with the D3 dispersion corrections proposed by Grimme et al.[52]. The surface/adsorbate system was modeled within the repeated slab scheme, i.e., a simulation cell containing 4 atomic layers of Au along the [111] direction and a layer of hydrogen atoms to passivate one side of the slab in order to suppress one of the two Au(111) surface states. Forty ångströms of vacuum were included in the simulation cell to decouple the system from its periodic replicas in the direction perpendicular to the surface. We considered a supercell of 82.54 Å × 45.95 Å corresponding to 502 surface units. A finite 7-AGNR of length ~6 nm was adsorbed on the gold slab. To obtain the equilibrium geometry we kept the atomic positions of the bottom two layers of the slab fixed to the ideal bulk positions; all other atoms were relaxed until the forces were lower than 0.005 eV/Å. To obtain the electron probability densities shown in Figs. 2c, 2d, 3i, 3j, and S3b we extrapolated the electronic orbitals obtained from CP2K to the vacuum region in order to correct the inaccurate decay in vacuum of the charge density due to the localized basis[53]. The orbital decays shown in Supplementary Fig. 8 are obtained by convoluting the orbitals of a ~9-nm-long 7-AGNR with an $s$-wavefunction with $1/e$ decay length of 0.5 Å (ref. [9,54]). The orbitals are computed with a mean field Hubbard third nearest neighbor tight-binding model (hopping terms of 2.7, 0.1, and 0.4 eV and U term of 3 eV). Linear combinations of Slater-type $2p_z$ functions are used for the real-space representation of the tight-binding orbitals.

**Modeling differential conductance and THz rectification.** Following reference 55, the Bardeen model for one-dimensional tunnelling is expressed as,

$$I(V) = \frac{4\pi e^2}{\hbar} \int_0^{eV} \rho_{\text{tip}}(\varepsilon - eV)\rho_{\text{sample}}(\varepsilon)T(\varepsilon, V, z)d\varepsilon, \quad (2)$$

where $e$ is elementary charge, $\hbar$ is the reduced Planck constant, $V$ is the bias applied to the sample, $\rho_{\text{tip}}$ is the tip density of states, $\rho_{\text{sample}}$ is the sample density of states and $T(\varepsilon, V, z)$ is the transmission coefficient that is approximated as $T(z) \propto \exp[-2\kappa z]$, where $\kappa$ is the inverse decay constant. The differential conductance (d$I$/d$V$) at a constant height, $z_0$, can be approximated as

$$\frac{dI}{dV}(V) = \frac{4\pi e^2}{\hbar}\rho_{\text{tip}}(0)\rho_{\text{sample}}(eV)T(z_0), \quad (3)$$

where we assume that the contribution at the top of the integral ($\varepsilon = eV$) dominates and leads to an energy-independent transmission coefficient. For a metallic tip, the tip density of states can be considered as constant, leaving d$I$/d$V$ proportional to the sample density of states, $\frac{dI}{dV}(V) \propto \rho_{\text{sample}}(eV)$. Our model for the tunnel junction d$I$/d$V$ is represented by three components: the GNR with two different error functions representing band onsets at positive and negative bias and a Gaussian is representing the Au(111) surface state as given by

$$\frac{dI}{dV}(V) = A_+ \cdot \text{erf}(c_+[eV - E_+]) + A_- \cdot \text{erf}(c_-[eV - E_-]) + A_{\text{Au}} \cdot e^{-c_{\text{Au}}(eV - E_{\text{Au}})^2}, \quad (4)$$

where $c_+$, $c_-$, and $c_{\text{Au}}$ are coefficients that determine the shape of the function and $E_+$, $E_-$, and $E_{\text{Au}}$ are the positions of these features as determined by fits to the measured d$I$/d$V$ in Fig. 2a, while $A_+$, $A_-$, and $A_{\text{Au}}$ are free parameters that control the amplitude of each feature. The height-dependent behavior is then encoded into the model by multiplying each component in the d$I$/d$V$ expression by its own transmission coefficient, $T(z)$, with independent $\kappa$ parameters. The current–voltage characteristic of the junction is modelled as the integral of d$I$/d$V$, i.e. $I(V) = \int_0^{eV} \frac{dI}{dV}(\varepsilon)d\varepsilon$. The amplitude of each term in the d$I$/d$V$ expression is determined by a least-squares fit to experimental $Q_{\text{THz}}$–$E_{\text{THz,pk}}$ curves. The rectified charge is the integral of the instantaneous current, $Q_{\text{THz}} = \int i_{\text{THz}}(t)dt$, across the junction induced by the terahertz voltage transient, $V_{\text{THz}}(t)$. To simulate $Q_{\text{THz}}$–$E_{\text{THz,pk}}$ curves we map $V_{\text{THz}}(t)$ onto the conductance determined by the model d$I$/d$V$ where $V_{\text{THz}}(t)$ is modeled by a normalized waveform measured with PES, multiplied by a scalar coefficient, $V_{\text{THz}}(t) = V_{\text{pk}} \cdot V_0(t)$. The peak bias applied to the junction, $V_{\text{pk}}$, is a product of the field strength of the incident terahertz pulse and a voltage calibration factor that scales with field enhancement at the microscopic tip apex[17,21]. For height-dependent modelling we assume that the voltage calibration is constant at all tip-sample distances, unless a tip change is

experimentally observed. This assumption is consistent with the observations of references 11 and 14. Additionally, following reference 9, we estimate the tunnelling time between the substrate and GNR by extrapolating the results from reference 56. At the limit of zero dielectric spacer layers, we estimate a tunnelling time in the range of 1–10 fs. This range is corroborated by our THz-STM measurements, which can be explained by the model above without invoking multi-electron effects.

## Data availability
The data that support the plots within this paper and other findings of this study are available from the corresponding author upon reasonable request.

## Code availability
The tight-binding calculations were performed using a custom-made Python program available on the GitHub repository (https://github.com/eimrek/tb-mean-field-hubbard). The codes and simulation files that support the plots and data analysis within this paper are available from the corresponding author on reasonable request.

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

## Acknowledgements

The authors thank R. Loloee for technical support and R. Huber and A. Sinitskii for valuable discussions. This project was financially supported by the Office of Naval Research (Grants no. N00014-19-1-2051, N00014-18-1-2708), the Cowen Family Endowment, and the Swiss National Science Foundation (Grant no. 200020_182015). The Swiss National Supercomputing Centre (CSCS) under project ID s904 is acknowledged for computational resources.

## Author contributions

S.E.A. and V.J. prepared the samples. P.R. and R.F. provided the precursor molecules and consulted on the on-surface synthesis and STM characterization of GNRs. S.E.A., V.J., V.N.B. and T.L.C. constructed the experimental setup. S.E.A., V.J. and T.L.C. conducted the experiments with support from M.H. and S.L. The data was analyzed by S.E.A., V.J., and T.L.C. using a model developed by S.E.A., Y.W., N.E., V.J. and T.L.C. Density functional theory calculations were performed by C.A.P. and Q.S. The study was conceived and supervised by T.L.C. The manuscript was written by S.E.A., V.J., and T.L.C. with input from all authors.

## Competing interests

The authors declare no competing interests.
