## [Peer Review File · Nature Communications]

Lightwave-driven scanning tunnelling spectroscopy of atomically precise graphene nanoribbonsREVIEWER COMMENTS

Reviewer #1 (Remarks to the Author):

The manuscript entitled "Lightwave-driven scanning tunneling spectroscopy of atomically precise graphene nanoribbons" by S.E. Ammerman and coworkers presents the first study of THz-STM on graphene nanoribbons (GNR). Using THz-STM, they access the LDOS of GNRs at tip heights that are difficult to operate at in conventional STM. The work presents high-quality THz-STM data revealing sub-Ångstrom details of the electronic structure of a GNR which have so far not been experimentally accessible, which is impressive and certainly adds new value and insight into the emerging field of THz-STM.

The manuscript is clearly written and the figures and data are of good quality and described properly throughout the manuscript. The descriptions of experimental procedures are clear and complete and allow for reproducibility of the work. The theoretical modeling is described throughout the text, though the model description could be improved by adding some equations. Also, some of the model assumptions should be better justified.

The main result of the manuscript is the demonstration of experimental access to the Ångstrom-resolved LDOS at very small gap distances by THz-STM. The obtained results are consistent with DFT calculations and previous experimental work. Although the work lacks new physical insight beyond confirmation of DFT predictions and enhanced application and modelling of established THz-STM concepts, it nevertheless denotes a major achievement in the development of lightwave-STM and demonstration of a new operation regime of STM. As this will be applicable and extendable to a variety of sample systems, and as it paves the way for future femtosecond atomic-scale spectroscopy, I consider it of high relevance and of interest to a broad audience.

I thus recommend publication of the manuscript in Nature Communication after the raised questions and points have been satisfactorily addressed.

I have several questions that should be addressed by the authors:

1) The drawn equivalence of THz-STs to static STS is somewhat confusing. In STS, the LDOS is directly measured as a function of energy (the bias modulation amplitude is fixed and the DC bias is swept). In contrast, the THz measurements shown in Fig. 3c-d measure an energy-integrated current as function of peak THz bias (the DC equivalent would be a sinusoidal bias modulation with increasing amplitude at 0 V DC). An alternative (and maybe the more intuitive) way to think of THz-STs would be to use a small THz amplitude modulating the bias around a DC bias that is swept, as in standard STS. This mode would be relevant for time-resolved studies yielding the transient LDOS and its transient occupation vs. energy. A discussion about these two approaches/understandings of THz-STs and the respective accessible information would add value to the manuscript and is desired to clarify the terminology 'THz-STs'.

2) To what extent are the THz-STs results unique over standard STS and how problematic is damaging of the ribbon? Other work, such as Koch et al. Nature Nanotechnology vol. 7, 713–717 (2012) could measure the conductance of a GNR after contact with tip over a large bias and current range without any damage. Is damage really the problem, or is it imaging and the limited spatial resolution at those vanishing gap distances and high current conditions?

3) THz-STs operation is (in part) in a regime of >1 electrons/pulse rectified current (especially at close distanced, Fig. 4a and 4b). Does multi-electron tunneling play a role? Is it clear and is there any evidence that each rectified electron tunnels into the neutral unexcited ribbon, and that bleaching effects due to a finite decay time into (or refilling time out of) the substrate are negligible? Has it been tested that the imaged LDOS distribution is independent on rectified charge?

4) In Fig. 1d) and in line 80 it is argued that peak currents in the μA -regime, as achieved during the peak of the THz pulse at ultralow tip heights, are non-destructive. I assume such destructive perturbation, but also transient heating and/or charging, will depend sensitively on the electron

density tunnelling within a certain time window during a half-cycle in relation to the equilibration time (decay rate of the transmitted charges) of the ribbon after single-electron excitation. Comparing the number of the transient current density and tunneling rate to the expected equilibration times should be provided to exclude non-equilibrium effects such as transient charging and heating (does the THz-signal probe the ribbon in its ground state?) and to corroborate the statement of non-destructive tunneling.

5) At the set point distance, the QTHz-ETHz,pk curves could be calculated directly from the DC I-V curve and the known calibrated THz voltage amplitude (see point 7). This would allow quantitative comparison of the measured and calculated rectified charge without any modelling. Is the QTHz-ETHz,pk curve calculated from a measured DC I-V curve and THz bias consistent with the measured one? Did the authors do such a self-consistence check?

If this is not possible at $z=z_0$ (due to low THz signal?), is there an intermediate gap distance at which a DC I-V curve can be recorded while a measurable rectified charge is detected? I recommend such self-consistence check to validate and demonstrate the retrieval of dI/dV information via the THz pulse without the need for any model assumptions.

6) The measured QTHz-ETHz,pk curves (Fig. 3c-d) contains structure and modulations that are not reproduced from the step-like error function used to model the LDOS (which are rough approximations anyways considering the modulations of the dI/dV curve at higher biases and the potentially large THz bias window). What is the origin of this structure, why can it not be reproduced by the model, and why is this not observed for the data set shown in Fig S5? Can THz-tunneling into higher states be excluded, or could this contribute to the measured signal and explain the modulations in Fig. 3c and 3d?

7) How large is the energy (bias) window sampled in the THz-STs data? What is the applied peak THz voltage obtained from the slope in Fig. S3b? The author use photoemission from the tip to measure the THz near-field waveform. Previous work has shown that this also allows calibration of the THz voltage. Why do the authors not use the calibrated THz voltage amplitude for data analysis and interpretation throughout the manuscript? Plotting the charge versus THz bias would be far more useful than plotting versus electric field.

Knowledge of the THz voltage amplitude would allow to judge what part of the LDOS is sampled by the THz bias, and would help to understand the appearance of spurious features in Fig. 3c (does the appearance of those correlate with the onset of accessing features in the I-V curve?). Does the calibrated THz voltage used corroborate the statement in line 132 that at $ETHz,pk > 0$ the THz bias probes considerable part of the LDOS on both sides of the Fermi level?

8) Is the THz voltage amplitude a free fit parameter, or is it fixed according to the known bias? Are the amplitudes of the three LDOS contributions the only free fit parameters? What justifies to keep the band edge positions and width constant versus tip height? Could it be possible that the band edge positions and width change with distances and not only their relative amplitudes?

9) Does the THz voltage and waveform remain constant at the ultralow tip heights? Is it possible that the significantly increased overlap of electron wave functions and tunneling probability at such small gap distances affects the THz voltage? It is known from plasmonic junctions that charge transfer across the gap can quench plasmonic enhancement in the visible at small gap distances. Can such quenching of the field enhancement also occur in the THz spectral range? As the modelling and the key results rely on the assumption that the THz waveform and its amplitude remain unaffected even at ultrashort gap distance, the authors should verify or argue why this assumption is justified.

10) Comparing the data sets from the two tips (Fig. 3 and Fig. S5), the QTHz-ETHz,pk curves of the tip used in Fig S5 shows less modulation and better fits. Is the modulated part of the QTHz-ETHz,pk curves in Fig. 3 that is not fitted well a feature of the tip, THz or the ribbon?

11) Following the discussion in Ref. 25, it is expected for 7-atom GNR that the CB decays faster with gap distance than the VB edge, opposite to what is retrieved from THz-STs data shown in Fig.

4. Why is this not observed? As the authors claim to retrieve the vertical decay of the LDOS, they should clarify this discrepancy and provide a more detailed and quantitative discussion explaining their contradictory observation. Are the slightly different barrier heights at the VB and CB, as argued in line 184, sufficient to explain the observed decay with opposite behavior than that expected from the wave function decay (ref. 25)?

12) Plotting line cuts along z through the maps in Fig. 4c-d would enhance clarity of the figure.

13) Fig. 1 should highlight more the novelty of the work by graphical illustration of the vertical decay or tomography of the states. Fig. 1d looks very similar to previous sketches on THz-STM operation and does not clearly deliver the novelty of the work and the used concept to access dI/dV information by THz-STM (it rather illustrates rectification).

14) Can the authors explain what is the difference between 'tomography' and 'z-spectroscopy'?

Reviewer #2 (Remarks to the Author):

S. E. Ammerman et al. report interesting results of lightwave-driven scanning tunnelling microscopy (STM), which reveals detailed structure of the wavefunctions by slightly approach the STM tip to the sample from the height normally used for conventional STM. The obtained images of the graphene nanoribbon are in excellent agreement with the calculated distribution of the electronic density with DFT method, indicating extremely high spatial resolution of the system. Since the electron distribution is of course very important for understanding electronic states of the materials, the reported new method could be a promising and useful tool for the surface science and materials science, and thus is of interest for the broad range of readers. Furthermore, this work reports a new perspective for the THz-STM system to use an ultralow tip height, which could open new applications of lightwave (or THz) driven STM, most probably by the combination with ultrafast time-resolution. The referee, therefore, thinks that the manuscript is worth publishing in Nature Communications when the following minor points are revised and clarified.

1. In the first paragraph, authors suggest the importance of time-resolved measurement. In the referee's understanding, the temporal resolution of the system, however, is not an important point of the manuscript. I think the temporal resolution is one of the prospects of the work, and the introduction should directly emphasize the important aspect of the manuscript, namely the sub-atomic imaging of the electron distribution inside the molecule.

2. The reason why the tunneling current will be proportional to the electronic density may better be validated with some equations as this will be important for understanding the mechanism how the tunneling current is determined. Especially the clear observation of the node in the obtained image might be interesting because this implies an interference of tunnelling pathways from the sample to the tip or vice a versa.

3. In Fig. 4, authors investigated the orbital decays by scanning the position of the tip. It may be meaningful if the authors quantitatively describe whether these values are consistent with the estimated values obtained by the DFT calculation or some other physical consideration (i.e. tunneling barrier height etc.).

4. In the experiment, CEP is observed to be sinusoidal rather than co-sinusoidal. Therefore, the asymmetry of the observed phenomena mainly originates from the asymmetry of the IV curve. The referee here come up with the question whether the co-sinusoidal waveform merit the interpretation of the experimental results further or not. Some additional comments on the role of terahertz waveform could be helpful.

5. Did the authors measure the STS map and can they map the fitting parameters of STS such as the amplitude of the step function as an image? It might be the next step, but it may help understand the origin of the observed images.

Reviewer #3 (Remarks to the Author):

In this work, the authors utilize THz-STM/STS to investigate seven-atom-wide armchair graphene nanoribbons. The THz-STM constant-height rectified charge maps reveal Angstrom resolution electronic wavefunction features of the nanoribbon. The authors also show that the vertical decay of the local density of states is sensitive to lateral position and electronic orbital. Although the results are interesting, this manuscript does not provide any new insights about electronic properties of armchair graphene nanoribbons. In fact, armchair graphene nanoribbons are the first type of graphene nanoribbons that were successfully synthesized (Nature 466, 470) and their electronic properties have been well investigated by STM/STS and Q-plus AFM (for instance, ACS Nano 6, 6930; PRB 91, 045429; Nature 560, 204). Therefore, I do not suggest to accept this manuscript for publication in Nature Communications.

Replies to Reviewers – Reviewer #1

Reviewer #1:

The manuscript entitled "Lightwave-driven scanning tunneling spectroscopy of atomically precise graphene nanoribbons" by S.E. Ammerman and coworkers presents the first study of THz-STM on graphene nanoribbons (GNR). Using THz-STM, they access the LDOS of GNRs at tip heights that are difficult to operate at in conventional STM. The work presents high-quality THz-STM data revealing sub-Ångstrom details of the electronic structure of a GNR which have so far not been experimentally accessible, which is impressive and certainly adds new value and insight into the emerging field of THz-STM.

The manuscript is clearly written and the figures and data are of good quality and described properly throughout the manuscript. The descriptions of experimental procedures are clear and complete and allow for reproducibility of the work. The theoretical modeling is described throughout the text, though the model description could be improved by adding some equations. Also, some of the model assumptions should be better justified.

The main result of the manuscript is the demonstration of experimental access to the Ångstrom-resolved LDOS at very small gap distances by THz-STM. The obtained results are consistent with DFT calculations and previous experimental work. Although the work lacks new physical insight beyond confirmation of DFT predictions and enhanced application and modelling of established THz-STM concepts, it nevertheless denotes a major achievement in the development of lightwave-STM and demonstration of a new operation regime of STM. As this will be applicable and extendable to a variety of sample systems, and as it paves the way for future femtosecond atomic-scale spectroscopy, I consider it of high relevance and of interest to a broad audience.

I thus recommend publication of the manuscript in Nature Communication after the raised questions and points have been satisfactorily addressed.

Reply #1.0:

We thank the Reviewer for their time and effort, their insightful and constructive criticism, and their positive evaluation of our work. Motivated by the Reviewers' comments, we have added new data (Fig. S5: Change #42) and analysis (Fig. 3g-j: Changes #13, #27, #28; Fig. S8: Changes #19, #32, #36, #43; Fig. S9: Changes #22, #44) to the manuscript. These additions strongly support our model. We have also added more description and justification of the model assumptions throughout the text (e.g. Changes #9, #12, #14, #15) and a new Methods section that details the model, rectification process, and relationship between differential conductance and local density of states (Changes #33, #37). We have also added a paragraph to the text that explains the current regime that THz-STM and THz-STs operate in and why this makes our new approach to THz-STs analysis essential (Change #6). We now include a plot that illustrates these concepts as Fig. S2 as well (Change #40). Collectively, these changes make the paper stronger and establish a new standard for THz-STM and THz-STs analysis.

Reviewer #1:

I have several questions that should be addressed by the authors:

1) The drawn equivalence of THz-STs to static STs is somewhat confusing. In STs, the LDOS is directly measured as a function of energy (the bias modulation amplitude is fixed and the DC bias is swept). In contrast, the THz measurements shown in Fig. 3c-d measure an energy-integrated current as function of peak THz bias (the DC equivalent would be a sinusoidal bias modulation with increasing amplitude at 0 V DC). An alternative (and maybe the more intuitive) way to think of THz-STs would be to use a small THz amplitude modulating the bias around a DC bias that is swept, as in standard STs. This mode would be relevant for time-resolved studies yielding the transient LDOS and its transient occupation vs. energy. A discussion about these two approaches/understandings of THz-STs and the respective accessible information would add value to the manuscript and is desired to clarify the terminology 'THz-STs'.

Reply #1.1:

The Reviewer's comments have alerted us that in the previous version of the manuscript, we did not sufficiently convey why our approach to THz-STs is essential. We agree that the Reviewer's suggested mode of operation is, in principle, more closely analogous to conventional STs. However, it is incompatible with lightwave-driven tunneling and ultrafast time resolution from a practical perspective because of the extremely low duty cycle of these experiments. We believe this is a critical point and therefore have added the following paragraph to the main text (Change #6) along with notes of further emphasis (Changes #8, #25).

Change #6

"In contrast [to conventional STs], in THz-STM and THz-STs, the terahertz voltage probe is nine orders of magnitude faster than conventional voltage modulation and only a single oscillation cycle long. As the terahertz pulses are generated by an ultrafast near-infrared laser (see Methods and Fig. S1), a train of terahertz pulses is delivered to the junction. Each terahertz pulse stimulates a sub-picosecond rectified current pulse through lightwave-driven tunneling, thereby producing a train of current pulses with a duty cycle equal to the current-pulse duration divided by the time between terahertz pulses ($\sim 1 \text{ ps} / 1 \text{ } \mu\text{s} = 10^{-6}$). Only the average terahertz-induced current can be measured, so the peak current must compensate for the duty cycle by far exceeding $I_{d.c.}$, as summarized in Fig. S2. Consequently, the conventional method of STs is incompatible with the key aspect of lightwave-driven tunneling that makes ultrafast time resolution possible, and an alternate approach is required for THz-STs."

We have also included Fig. S2 (Change #40) on the next page for ease of reference. I_{peak} refers to the peak terahertz-induced current in a 400-fs current pulse. I_{avg} is the average current that can be detected by the STM electronics, which have low bandwidth and therefore average over the pulse train. The shaded box in red shows the typical tunnel current detection limit of conventional STM systems that utilize lock-in detection ($\sim 10 \text{ fA}$), while the shaded box in yellow shows the typical range of the tunnel current for a high gain preamplifier operating in constant-current mode (active feedback loop). The dark shaded box on the right highlights a region that requires new technological advancements to achieve the necessary terahertz electric fields to drive tunneling across the junction. We estimate that the mode of operation suggested by the Reviewer would require laser repetition rates in the dark shaded region on the right-hand-side of the plot.

Reviewer #1:

2) To what extent are the THz-STs results unique over standard STS and how problematic is damaging of the ribbon? Other work, such as Koch et al. Nature Nanotechnology vol. 7, 713–717 (2012) could measure the conductance of a GNR after contact with tip over a large bias and current range without any damage. Is damage really the problem, or is it imaging and the limited spatial resolution at those vanishing gap distances and high current conditions?

Reply #1.2:

Imaging is indeed the key distinction. We demonstrate THz-STs with ångström (or better) spatial resolution in all three dimensions, at ultralow tip heights. Conventional spatially dependent STS is not feasible at these tip heights due to the high average tunnel currents that would be needed. Meanwhile, the paper by Koch *et al.* reports on contact measurements in which the tip picks up the GNR, making imaging impossible. However, we agree that our explanation of why conventional imaging is unfeasible at ultralow tip heights is incomplete. We have modified the corresponding statement to include not only damage to the tip and sample as possibilities, but also the tip picking up the GNR (Change #7) and have added the paper by Koch *et al.* to the reference list (Change #34). We have also modified the imprecise “non-destructive” description of terahertz-induced tunneling (Change #7).

Reviewer #1:

3) THz-STs operation is (in part) in a regime of >1 electrons/pulse rectified current (especially at close distanced, Fig. 4a and 4b). Does multi-electron tunneling play a role? Is it clear and is there any evidence that each rectified electron tunnels into the neutral unexcited ribbon, and that bleaching effects due to a finite decay time into (or

refilling time out of) the substrate are negligible? Has it been tested that the imaged LDOS distribution is independent on rectified charge?

Reply #1.3:

This is a fascinating point, and one we also considered closely, since the possibility of studying multi-electron effects in the time domain is a spectacular prospect for lightwave-driven STM. However, unfortunately, we do not find evidence for such effects in GNRs on gold. We have searched for any breakdown in our model due to multi-electron effects through field-dependent THz-STM imaging of the GNR. We have added this data to the manuscript (Fig. S5, Change #42) along with analysis in which we construct a $Q_{\text{THz}}-E_{\text{THz,pk}}$ curve at every pixel and then extract two-dimensional dI/dV maps at positive and negative bias (Fig. 3g,h, Changes #27, #28). No change in terahertz-induced tunneling behavior is observed as Q_{THz} increases from 2 electrons per pulse to 10 electrons per pulse, and the model produces maps that agree well with DFT predictions (Fig. 3i,j, Changes #27, #28). In describing the results of this new dataset in the text (Change #13), we have also expanded the explanation of the autocorrelation measurements from the original manuscript (Change #14), since these are an even better test of multi-electron and dynamic effects. A single dI/dV and the measured photoemission waveform are sufficient to explain the autocorrelation data at each tip position, meaning multi-electron effects must play a minor role. We suspect a dielectric spacer layer will be needed to observe them, and we have added a comment to this effect in the Discussion (Changes #23, #35).

Reviewer #1:

4) In Fig. 1d) and in line 80 it is argued that peak currents in the μA -regime, as achieved during the peak of the THz pulse at ultralow tip heights, are non-destructive. I assume such destructive perturbation, but also transient heating and/or charging, will depend sensitively on the electron density tunnelling within a certain time window during a half-cycle in relation to the equilibration time (decay rate of the transmitted charges) of the ribbon after single-electron excitation. Comparing the number of the transient current density and tunneling rate to the expected equilibration times should be provided to exclude non-equilibrium effects such as transient charging and heating (does the THz-signal probe the ribbon in its ground state?) and to corroborate the statement of non-destructive tunneling.

Reply #1.4:

Tunnelling time between a van der Waals bonded molecule (e.g. a GNR) and a substrate is notoriously difficult to calculate. In reference 9, it is argued based on the data in reference 56 that the tunnel time from the substrate to a molecule through a monolayer of NaCl is on the order of 100 fs. Extending this extrapolation further, we find a rough estimate of 1 - 10 fs for the tunnel time in the absence of a salt barrier. This is likely to be an upper bound since molecular orbitals can hybridize more strongly with the substrate in the absence of a dielectric spacer. We have added this estimate to the Methods (Changes #33, #38). The estimate is consistent with our THz-STM and THz-STS data and analysis, which can be explained by our model without invoking multi-electron effects. We have further added a comment in the main text that our autocorrelations serve as a test for multi-electron effects (Change #14) and another comment that a decoupling layer will likely be needed to observe the ultrafast tunnelling dynamics of GNRs in the future (Change #23).

Reviewer #1:

5) At the set point distance, the $Q_{\text{THz}}\text{-}E_{\text{THz,pk}}$ curves could be calculated directly from the DC I-V curve and the known calibrated THz voltage amplitude (see point 7). This would allow quantitative comparison of the measured and calculated rectified charge without any modelling. Is the $Q_{\text{THz}}\text{-}E_{\text{THz,pk}}$ curve calculated from a measured DC I-V curve and THz bias consistent with the measured one? Did the authors do such a self-consistence check?

If this is not possible at $z=z_0$ (due to low THz signal?), is there an intermediate gap distance at which a DC I-V curve can be recorded while a measurable rectified charge is detected? I recommend such self-consistence check to validate and demonstrate the retrieval of dI/dV information via the THz pulse without the need for any model assumptions.

Reply #1.5:

As explained in Reply #1.1, the peak terahertz-induced current needs to exceed $I_{\text{d.c.}}$ by many orders of magnitude in order to result in a measurable signal. In our experiments, we observe a Q_{THz} that is large enough to perform THz-STs for tip heights of $z = z_0 - 3 \text{ \AA}$ or closer. At these tip heights, the DC bias cannot be swept to the band onsets without disturbing the system (e.g. damaging the tip or moving the GNR). Therefore, such a crosscheck is not feasible for GNRs. That is why we use the STS differential conductance at z_0 as the point of comparison for the THz-STs model at lower tip heights. On the other hand, comparisons like the one the Reviewer suggests have been performed extensively in the literature for solids (see, e.g., reference 16), where the concept of charge rectification by lightwave-driven tunnelling has been consistently confirmed. We address the terahertz voltage calibration in Reply #1.7.

Reviewer #1:

6) The measured $Q_{\text{THz}}\text{-}E_{\text{THz,pk}}$ curves (Fig. 3c-d) contains structure and modulations that are not reproduced from the step-like error function used to model the LDOS (which are rough approximations anyways considering the modulations of the dI/dV curve at higher biases and the potentially large THz bias window). What is the origin of this structure, why can it not be reproduced by the model, and why is this not observed for the data set shown in Fig S5? Can THz-tunneling into higher states be excluded, or could this contribute to the measured signal and explain the modulations in Fig. 3c and 3d?

Reply #1.6:

The model we employ reproduces the consistent features observed across the two datasets in Fig. 3c,d and Fig. 7c,d. Introducing further parameters and terms into the model would be unnecessary for Fig. 7c,d, and therefore we opted for the model with the least number of free parameters. As the field develops, increasingly finer experimental details will be compared to theoretical predictions, but it is important to first establish the groundwork of spatially dependent THz-STs.

To expand on the experimental details, the dataset in Fig. S7 was recorded on a different day, with a different tip apex, and over a different GNR than the THz-STM and THz-STs data in the main text. The conventional STS in Fig. 3a is also an example of a typical spectrum and not specific to either dataset. Both the configuration of the GNR end states and the GNR's proximity to neighboring structures can affect the fine details of the GNR's electronic structure, but neither were controlled for in this study. Variation in the tip density of states may also have played a role, e.g. changes to the tip apex are routinely observed to affect the appearance of the gold surface state in STS. Meanwhile, tunneling into higher states is possible, but the spatial structure of our THz-STM images is clearly dominated by the band onsets. For example, the dI/dV maps extracted from the new field-dependent THz-STM imaging dataset match well to the DFT-predicted electron density at the band onsets (Fig. 3g-j, Changes #13, #27, #28).

Reviewer #1:

7) How large is the energy (bias) window sampled in the THz-STs data? What is the applied peak THz voltage obtained from the slope in Fig. S3b? The author use photoemission from the tip to measure the THz near-field waveform. Previous work has shown that this also allows calibration of the THz voltage. Why do the authors not use the calibrated THz voltage amplitude for data analysis and interpretation throughout the manuscript? Plotting the charge versus THz bias would be far more useful than plotting versus electric field.

Knowledge of the THz voltage amplitude would allow to judge what part of the LDOS is sampled by the THz bias, and would help to understand the appearance of spurious features in Fig. 3c (does the appearance of those correlate with the onset of accessing features in the I-V curve?). Does the calibrated THz voltage used corroborate the statement in line 132 that at $E_{\text{THz},pk} > 0$ the THz bias probes considerable part of the LDOS on both sides of the Fermi level?

Reply #1.7:

THz-STM is a developing technique, and therefore some practical considerations have not yet been fully detailed in the literature. Although photoemission sampling is a valuable tool for characterizing the temporal shape of the waveform at the tip apex, it does not necessarily provide a reliable voltage calibration for THz-STM imaging. We feel this is an important point and therefore we have added the following explanation to the main text.

Change #9

“As terahertz field enhancement is affected by the microscopic tip profile^{17,21} – which must be carefully prepared over the substrate for high-resolution THz-STM imaging – we use PES to measure the waveform shape but determine the terahertz voltage calibration from THz-STM and THz-STs.”

As an additional note, references 17 and 21 reported that the field-enhancement factor can change when the atomic configuration of the tip apex is modified, which is consistent with our experimental observations. The new field-dependent imaging and analysis (Changes #13, #27, #28, #42) help to clarify the voltage range probed, since the extracted dI/dV maps can be compared directly to DFT calculations. This comparison indicates that terahertz-induced tunneling is dominated by the calculated LDOS features observed at -0.9 V and at +1.3 V.

Reviewer #1:

8) Is the THz voltage amplitude a free fit parameter, or is it fixed according to the known bias? Are the amplitudes of the three LDOS contributions the only free fit parameters? What justifies to keep the band edge positions and width constant versus tip height? Could it be possible that the band edge positions and width change with distances and not only their relative amplitudes?

Reply #1.8:

The terahertz field enhancement, which determines the voltage amplitude, is a fit parameter, but it is kept constant unless a tip change is observed experimentally. We have included a number of statements throughout the text clarifying the role of the field enhancement as a fit parameter (Changes #9, #12, #13, #14, #33).

It is possible the band edges or widths could change. However, we tested the model with these additional free parameters and did not see substantial improvement to the THz-STs fits. Therefore, we do not have any evidence that it occurs. We acknowledge our model is simple, yet it is sufficient to explain almost all our observations with only a few fit parameters and without invoking more complicated effects.

Reviewer #1:

9) Does the THz voltage and waveform remain constant at the ultralow tip heights? Is it possible that the significantly increased overlap of electron wave functions and tunneling probability at such small gap distances affects the THz voltage? It is known from plasmonic junctions that charge transfer across the gap can quench plasmonic enhancement in the visible at small gap distances. Can such quenching of the field enhancement also occur in the THz spectral range? As the modelling and the key results rely on the assumption that the THz waveform and its amplitude remain unaffected even at ultrashort gap distance, the authors should verify or argue why this assumption is justified.

Reply #1.9:

The THz-STM autocorrelation measurements (Fig. S6) serve as a test for this effect. We find that our model, combined with the measured photoemission waveform, sufficiently captures the details of the field- and position-dependent autocorrelations. If the photoemission waveform were modified at small gap distances, this would not be the case. We have expanded the discussion of the autocorrelation measurements in the manuscript and added a comment that they serve to corroborate our model and the photoemission waveform, in addition to acting as a test for more complex effects (Change #14).

Regarding the broader possibility that the terahertz voltage waveform could change as a function of tip position, tunnel current, or peak voltage, we agree that this could happen for measurement regimes beyond what we explore in the current manuscript as well as in other systems in which terahertz resonances are present. The terahertz field enhancement at the tip apex can be thought of as the result of a capacitor with an extremely small gap size. Shorting the capacitor would, in principle, reduce the field enhancement, as would charge accumulation, which is noted in

reference 17. However, we do not see evidence for this in our samples, in the regimes we have explored. It remains an interesting future prospect to explore that, like dynamics and multi-electron effects, we may require a dielectric spacer layer to see. Finally, we believe it is worth noting that the terahertz frequency is approximately three orders of magnitude below the plasma frequency of the tip, so the physics governing its field enhancement is substantially different (please see J. Appl. Phys. **128**, 070901 (2020) for further details).

Reviewer #1:

10) Comparing the data sets from the two tips (Fig. 3 and Fig. S5), the QTHz-ETHz,pk curves of the tip used in Fig S5 shows less modulation and better fits. Is the modulated part of the QTHz-ETHz,pk curves in Fig. 3 that is not fitted well a feature of the tip, THz or the ribbon?

Reply #1.10:

The tip density of states and the ribbon's configuration in terms of end states, proximity to other ribbons, and adsorption site could play a role. It is unlikely that the difference is due to the terahertz waveform or voltage because the macroscopic and mesoscopic tip shape are identical (so the photoemission waveform applies to both datasets) while field enhancement changes due to the microscopic apex are accounted for by the terahertz voltage calibration. We speculate that the tip density of states plays the dominant role, as this would be consistent with our observation that the precise shape of the differential conductance measured by steady-state STS depends on the microscopic tip apex.

Reviewer #1:

11) Following the discussion in Ref. 25, it is expected for 7-atom GNR that the CB decays faster with gap distance than the VB edge, opposite to what is retrieved from THz-STs data shown in Fig. 4. Why is this not observed? As the authors claim to retrieve the vertical decay of the LDOS, they should clarify this discrepancy and provide a more detailed and quantitative discussion explaining their contradictory observation. Are the slightly different barrier heights at the VB and CB, as argued in line 184, sufficient to explain the observed decay with opposite behavior than that expected from the wave function decay (ref. 25)?

Reply #1.11:

We apologize, this is an issue of imprecise language in the previous version of the manuscript, not a contradiction with the results of reference 26 (previously reference 25). The conduction band (CB) and CB + 1 are closely spaced in energy (see reference 26). Our experiments are performed at liquid nitrogen temperatures, and therefore these bands are indistinguishable in spectroscopy. We use a single error function to describe the total density of states at the positive band onset in our model, but this can be understood to include both the CB and CB + 1. However, their decays may differ.

Unfortunately, we cannot compare the vertical decays of differential conductance extracted from lightwave-driven measurements to the DFT calculations (which include both CB and CB + 1) because the vertical LDOS decays predicted by DFT are not reliable due to the fast decay of the Gaussian basis set and limitations of the exchange correlation functional, which we now note in

the text (Change #19). Therefore, in the revised manuscript, we introduce new tight-binding calculations that show the vertical decay of each orbital individually (VB - 1, VB, CB, CB + 1) and compare these to cuts through Fig. 4c,d (Fig. S8, Changes #19, #32, #36, #43). As we now explain in the text (Change #19), reasonable agreement is obtained, albeit under the assumption that each experimental decay can be associated with a particular band that dominates at that position and energy. In the positive bias case, it appears we observe mainly the decay of the CB + 1 in our experiments.

We have further corrected our imprecise language describing the band onsets as solely due to the VB and CB in the text (Changes #1, #3, #4, #5, #10, #11, #16, #17, #18, #21, #26, #29).

Reviewer #1:

12) Plotting line cuts along z through the maps in Fig. 4c-d would enhance clarity of the figure.

Reply #1.12:

Line cuts have been added as Fig. S8 alongside theoretical decays for each band (VB - 1, VB, CB, CB + 1) calculated by tight-binding (Changes #19, #32, #36, #43).

Reviewer #1:

13) Fig .1 should highlight more the novelty of the work by graphical illustration of the vertical decay or tomography of the states. Fig. 1d looks very similar to previous sketches on THz-STM operation and does not clearly deliver the novelty of the work and the used concept to access dI/dV information by THz-STM (it rather illustrates rectification).

Reply #1.13:

We have added three-dimensional renderings of the DFT images (at negative bias) to Fig. 1a,b that now illustrate how new information can be revealed at THz-STM tip heights (Changes #24, #25). Reply #1.1 explains why the comparison between Fig. 1c and 1d is essential for understanding THz-STS. This is made clearer in the manuscript through the addition of a new paragraph highlighting the differences between STS and THz-STS (Change #6) as well as the new Fig. S2 that illustrates the practical aspects that must be considered for a THz-STS experiment (Change #40).

Reviewer #1:

14) Can the authors explain what is the difference between 'tomography' and 'z-spectroscopy'?

Reply #1.14:

The distinction between THz-STM z-spectroscopy, which measures $Q_{\text{THz}}(z)$, and THz-STM tomography, which measures $Q_{\text{THz}}(x,y)$ at a sequence of z positions, has been made more explicit in the text (Changes #20, #22, #29). Furthermore, our tomography terminology is even more

appropriate in the revised manuscript, as it now features tomographic reconstructions of $d//dV$ as a function of x and y at a sequence of z settings (Fig. S9, Changes #22, #44).

Reviewer #2:

S. E. Ammerman et al. report interesting results of lightwave-driven scanning tunneling microscopy (STM), which reveals detailed structure of the wavefunctions by slightly approach the STM tip to the sample from the height normally used for conventional STM. The obtained images of the graphene nanoribbon are in excellent agreement with the calculated distribution of the electronic density with DFT method, indicating extremely high spatial resolution of the system. Since the electron distribution is of course very important for understanding electronic states of the materials, the reported new method could be a promising and useful tool for the surface science and materials science, and thus is of interest for the broad range of readers. Furthermore, this work reports a new perspective for the THz-STM system to use an ultralow tip height, which could open new applications of lightwave (or THz) driven STM, most probably by the combination with ultrafast time-resolution. The referee, therefore, thinks that the manuscript is worth publishing in Nature Communications when the following minor points are revised and clarified.

Reply #2.0:

We thank the Reviewer for their time and effort, their insightful and constructive criticism, and their positive evaluation of our work. Motivated by the Reviewers' comments, we have added new data (Fig. S5: Change #42) and analysis (Fig. 3g-j: Changes #13, #27, #28; Fig. S8: Changes #19, #32, #36, #43; Fig. S9: Changes #22, #44) to the manuscript. These additions strongly support our model. We have also added more description and justification of the model assumptions throughout the text (e.g. Changes #9, #12, #14, #15) and a new Methods section that details the model, rectification process, and relationship between differential conductance and local density of states (Changes #33, #37). We have also added a paragraph to the text that explains the current regime that THz-STM and THz-STs operate in and why this makes our new approach to THz-STs analysis essential (Change #6). We now include a plot that illustrates these concepts as Fig. S2 as well (Change #40). Collectively, these changes make the paper stronger and establish a new standard for THz-STM and THz-STs analysis.

Reviewer #2:

1. In the first paragraph, authors suggest the importance of time-resolved measurement. In the referee's understanding, the temporal resolution of the system, however, is not an important point of the manuscript. I think the temporal resolution is one of the prospects of the work, and the introduction should directly emphasize the important aspect of the manuscript, namely the sub-atomic imaging of the electron distribution inside the molecule.

Reply #2.1:

We have modified the introduction to highlight that, along with ultrafast time resolution, lightwave-driven microscopy also opens up the possibility of operating in regimes inaccessible to conventional STM (Change #2). This applies to ultralow tip heights, as is the case here, as well as extreme tunnel currents observed previously.

Reviewer #2:

2. The reason why the tunneling current will be proportional to the electronic density may better be validated with some equations as this will be important for understanding the mechanism how the tunneling current is determined. Especially the clear observation of the node in the obtained image might be interesting because this implies an interference of tunneling pathways from the sample to the tip or vice versa.

Reply #2.2:

We have added a new Methods section explaining, with equations, how the differential conductance in STM relates to the local density of states of a sample (Changes #33, #37). This new Methods section also shows the equations for terahertz-pulse rectification and our model (Change #33). We note that the observation of anti-nodes and nodes in the THz-STM images is related to the spatial distribution of the wavefunctions themselves (see Fig. S3) rather than interference between tunneling pathways. This point is made clearer by our new field-dependent imaging dataset (Fig. S5, Change #42), from which we extract experimental dI/dV maps (Fig. 3g,h, Changes #13, #27, #28) with spatial structure matching that of the DFT-calculated electron density at the band onsets (Fig. 3i,j, Changes #13, #27, #28).

Reviewer #2:

3. In Fig. 4, authors investigated the orbital decays by scanning the position of the tip. It may be meaningful if the authors quantitatively describe whether these values are consistent with the estimated values obtained by the DFT calculation or some other physical consideration (i.e. tunneling barrier height etc.).

Reply #2.3:

Unfortunately, whereas DFT calculations of the local density of states agree well with our THz-STM and THz-STS measurements (Fig. 3g-j, Changes #13, #27, #28), the vertical decays predicted by DFT are not reliable due to the fast decay of the Gaussian basis set and limitations of the exchange correlation functional. We have added a comment in the text to this effect (Change #19). Therefore, to address the Reviewer's comment fully, we have added a comparison of the measured decays to tight-binding calculations as Fig. S8 (Changes #32, #36, #43). As written in the text (Change #19), reasonable agreement is obtained, albeit under the assumption that each experimental decay can be associated with one of the closely spaced frontier bands.

Reviewer #2:

4. In the experiment, CEP is observed to be sinusoidal rather than co-sinusoidal. Therefore, the asymmetry of the observed phenomena mainly originates from the asymmetry of the IV curve. The referee here come up with the question whether the co-sinusoidal waveform merit the interpretation of the experimental results further or not. Some additional comments on the role of terahertz waveform could be helpful.

Reply #2.4:

The waveform used in the study is not a perfect sine shape, but rather has an asymmetry between the main peak amplitudes of approximately 20%. This is a key aspect of the current study because the asymmetry leads to different images when the strongest peak field is negative vs. positive (compare, for example, Fig. 2g and 2h). If the waveform were symmetric, this difference would not be observed. Increasing the asymmetry of the waveform (e.g. towards a cosine-like shape) would further accentuate the differences. However, our THz-STs model allows us to effectively deconvolute the bipolar tunneling process and extract the underlying differential conductance. This would be very difficult, if not impossible, with a perfectly symmetric waveform. We have added a comment to this effect to the Methods section (Change #30).

Reviewer #2:

5. Did the authors measure the STS map and can they map the fitting parameters of STS such as the amplitude of the step function as an image? It might be the next step, but it may help understand the origin of the observed images.

Reply #2.5:

We thank the reviewer for this fantastic suggestion! In the revised manuscript, we have added a field-dependent THz-STM imaging dataset (Fig. S5, Change #42), which allows us to combine THz-STM with THz-STs. As described in the text (Change #13), we assemble $Q_{\text{THz}}-E_{\text{THz},pk}$ curves for every pixel from this dataset and apply our model to each pixel individually. This yields extracted dI/dV maps for negative and positive bias voltage (Fig. 3g,h, Changes #27, #28) that we can compare directly to DFT calculations of the electron density at the band onsets (Fig. 3i,j, Changes #27, #28).

Inspired by the Reviewer's suggestion, we took this concept one step further. We assembled $Q_{\text{THz}}-z$ curves for every pixel of Fig. 4e,f, allowing us to apply our model as in Fig. 4a,b and extract dI/dV maps at negative and positive bias at each tip height (Fig. S9, Changes #22, #44), i.e. tomographic reconstructions. We were even able to extract spatial maps of the inverse decay constant, κ , for the positive and negative bands and find that these agree with our point-spectroscopy results (Changes #22, #44).

Reviewer #3:

In this work, the authors utilize THz-STM/STS to investigate seven-atom-wide armchair graphene nanoribbons. The THz-STM constant-height rectified charge maps reveal Angstrom resolution electronic wavefunction features of the nanoribbon. The authors also show that the vertical decay of the local density of states is sensitive to lateral position and electronic orbital. Although the results are interesting, this manuscript does not provide any new insights about electronic properties of armchair graphene nanoribbons. In fact, armchair graphene nanoribbons are the first type of graphene nanoribbons that were successfully synthesized (Nature 466, 470) and their electronic properties have been well investigated by STM/STS and Q-plus AFM (for instance, ACS Nano 6, 6930; PRB 91, 045429; Nature 560, 204). Therefore, I do not suggest to accept this manuscript for publication in Nature Communications.

Reply #3.0:

We respectfully disagree with the Reviewer's evaluation of the manuscript's novelty and highlight the summary statements written by Reviewers #1 and #2:

Reviewer #1 writes, "The work presents high-quality THz-STM data revealing sub-Ångstrom details of the electronic structure of a GNR which have so far not been experimentally accessible, which is impressive and certainly adds new value and insight into the emerging field of THz-STM," adding that the work "denotes a major achievement in the development of lightwave-STM and demonstration of a new operation regime of STM. As this will be applicable and extendable to a variety of sample systems, and as it paves the way for future femtosecond atomic-scale spectroscopy, I consider it of high relevance and of interest to a broad audience."

Meanwhile, Reviewer #2 writes, "...the reported new method could be a promising and useful tool for the surface science and materials science, and thus is of interest for the broad range of readers. Furthermore, this work reports a new perspective for the THz-STM system to use an ultralow tip height...."

REVIEWERS' COMMENTS

Reviewer #1 (Remarks to the Author):

Review to the revised manuscript entitled "Lightwave-driven scanning tunneling spectroscopy of atomically precise graphene nanoribbons" by S.E. Ammerman and coworkers.

The authors replied to all comments in very detail and provided conclusive explanations for the raised questions. The changes made to the manuscript are consistent, add additional value and enhance the readers understanding of the details of the research. Besides one minor suggestion, I recommend publication of the revised version of the manuscript in Nature Communication.

Response to Reply #1.1:

For the introduction of THz-STs and its comparison to standard STs, I recommend to write the statement of line 164, that for THz-STs the peak THz bias is swept over a large range at 0 V DC bias, in an earlier paragraph (it should be mentioned in the paragraph starting line 83). Whereas the low duty cycle is the actual reason for this implementation, the sentence in line 164 specifically describes how spectroscopic information is obtained in THz-STs and how this is technically realized. It will be helpful especially for non-expert readers to understand this at an early stage in the manuscript.

Reviewer #2 (Remarks to the Author):

Authors successfully respond to the referee's comments. The referee thinks the manuscript is of high quality and importance, and recommends publication in Nature Communications.